# Multi-View Graph Disentanglement via Joint Contrastive Optimization

## Abstract

Graph Representation Learning (GRL) has made great progress by optimizing node representations through constructing multiple views and employing mutual information maximization or contrastive learning methods. However, existing methods typically rely on graph augmentation to construct node- and graph-level views by maximizing inter-view consistency. This strategy tends to force different views toward homogeneity, and may discard critical information in the graph data. Meanwhile, views at different hierarchical levels exhibit inherent limitations: node-level views are sensitive to noise, and graph-level views overlook local structural information. In this work, we propose **M**ulti-view **G**raph **R**epresentation **L**earning with **D**isentanglement via joint contrastive optimization (MGDRL). Multi-view graph disentanglement (MGD) promotes divergence among representations across different views, forming decoupled views. However, disentanglement alone may lead to meaningless representations. Therefore, we employ fuzzy self-attention mechanism to construct an aggregation graph and achieve synergistic constraints between the aggregation graph and MGD through joint contrastive optimization. Joint contrastive optimization guides decoupled views toward distinct and diverse information while also extending contrastive learning to the subgraph-level of the aggregation graph, integrating local and global information. Experimental results on benchmark datasets demonstrate the superior performance of MGDRL.

## 1 Introduction

Graph Neural Networks (GNNs) propagate information between nodes through a message-passing mechanism, modeling local structural patterns, and have demonstrated exceptional performance in tasks such as node classification (Shen et al., 2025), link prediction (Cai et al., 2021), and graph classification (Zhang et al., 2018). Most existing GNNs are trained in a supervised manner, but they often rely on large amounts of labeled data, which limits their applicability in tasks such as protein function prediction (Zhang et al., 2024c) where expert annotations are scarce, and as network depth increases, they become prone to over-smoothing, reducing their discriminative ability (Li et al., 2018).

Graph Representation Learning (GRL) combines view-invariance learning with graph structural properties (Zhao et al., 2025), using self-supervised methods to extract salient information from the data and thereby reduce the model's dependence on labeled examples. As one of the core paradigms of GRL, Graph Contrastive Learning (GCL) aims to learn representations by constructing different views of samples and comparing positive and negative instances in the embedding space (Zhang et al., 2023; Li et al., 2023; Zhang et al., 2024a).

To enhance the generalization and robustness of representation learning on graph-structured data, GCL employs various graph augmentation techniques to generate augmented views (Ding et al., 2022), such as feature masking (Zhu et al., 2021; You et al., 2020), edge perturbationRong et al. (2020), node dropping (You et al., 2020), subgraph samplingQiu et al. (2020). After obtaining different views, most GCL methods learn discriminative and invariant essential representations in graph data by maximizing mutual information (Veličković et al., 2019). Mutual information maximization methods can be categorized into two types (Zhao et al., 2025). The first (Veličković et al., 2019; Peng et al., 2020; Zhao et al., 2023) constructs positive and negative sample pairs between

node representations and a global graph or a local summary. The second maximizes a lower bound on mutual information (Gutmann & Hyvärinen, 2010; Oord et al., 2018; Sohn, 2016). Despite the flourishing development of GCL, this paradigm still has some drawbacks.

Theoretical and empirical research both indicate that effective augmented views should exhibit diversity while preserving the integrity of task-relevant information (Tian et al., 2020; Gong et al., 2023). However, most existing GCL methods employ manual graph augmentation strategies (Ding et al., 2022), which cannot guarantee that task-relevant information is preserved and may even severely disrupt graph topologies highly related to downstream tasks, resulting in low-quality embeddings (Zhu et al., 2021).

On the other hand, existing GCL methods focus on generating node- and graph-level contrastive views (Zhao et al., 2025). Node-level views can effectively capture feature information for individual nodes. However, they are highly sensitive to feature perturbations and node noise (Ju et al., 2024). Graph-level views provide a global representation, but because they rely on global pooling, they struggle to preserve structural patterns such as functional modules or intra-community dependencies, resulting in ambiguity in local semantics (Ying et al., 2018). In addition, existing GCL methods treat corresponding nodes across different views as positive pairs (Shen et al., 2023), which drives the representations of the views to converge toward similarity and fails to fully exploit the multi-view capacity to capture diverse representations.

To obtain more diverse representations without applying graph perturbations and to overcome the limitations of contrastive views, we propose MGDRL, a graph representation learning framework that performs multi-view graph disentanglement (MGD) via joint contrastive optimization. In MGDRL, we employ MGD to generate three decoupled views of graph data, enabling diverse representations across the decoupled views without compromising critical graph information. However, the driving force provided by disentanglement alone is insufficient and may lead to meaningless representations. Therefore, we propose joint contrastive optimization, utilizing contrastive learning as a constraint for MGD to drive decoupled views to learn diverse semantic information. First, we aggregate the decoupled views through fuzzy self-attention to obtain the aggregation graph. Based on their distinct roles in the aggregation process, these decoupled views are designated as the *transition* ($t$), *readout* ($r$) and *embedding* ($e$) views, respectively. Then, we use the aggregation graph and decoupled view $e$ as contrastive objectives to perform node-subgraph and subgraph-subgraph level comparisons, which we refer to as subgraph contrastive learning. Using joint contrastive optimization not only encourages the decoupled views to focus on distinct and diverse information, but also extends the contrastive objective to subgraph-level views, thereby achieving a integration between local and global information. The main contributions of this work are as follows:

- We construct MGD to obtain decoupled views that capture diverse information without graph augmentation.
- We propose subgraph contrastive learning, which extends the contrastive objective to the node-subgraph and subgraph-subgraph levels, enabling the aggregation graph to integrate local and global information.
- We propose joint contrastive optimization, which uses fuzzy self-attention aggregation as a bridge between MGD and subgraph contrastive learning, to obtain diverse information across decoupled views and comprehensive node- and graph-level representations in the aggregation graph.
- Experimental results on multiple datasets show that MGDRL performs excellently on semi-supervised node classification and node clustering tasks, even outperforming some supervised GNNs.

## 2 RELATED WORK

**Graph Augmentation** In GRL, before performing contrastive learning, the original graph is augmented in various ways to obtain multiple augmented views. For example, GraphCL (You et al., 2020) systematically introduced four basic augmentation operations, namely feature masking (Zhu et al., 2021; You et al., 2020), edge perturbation (Rong et al., 2020), node dropping (You et al., 2020) and subgraph sampling (Qiu et al., 2020), to construct augmented views. GRACE (Zhu et al., 2020) and CCA-SSG (Zhang et al., 2021) use random edge perturbation and feature masking to

generate two views. MVGRL (Hassani & Khasahmadi, 2020) constructs augmented views using graph diffusion. SubgDiff (Zhang et al., 2024b) incorporates molecular subgraph information into diffusion to enhance the awareness of the denoising network of molecular substructures. To better align graph augmentations with downstream tasks, adaptive augmentation methods have gradually emerged. GCA (Zhu et al., 2021) applies selective augmentations based on graph structure and node feature importance, assigning higher masking probability to less important elements. AutoGCL (Yin et al., 2022) adjusts perturbation strength dynamically to balance diversity and the integrity of task-relevant information. GOUDA (Zhuo et al., 2024) employs a learnable unified graph augmentation module to simulate arbitrary explicit graph augmentations. To avoid information loss and reduced generalization from handcrafted augmentations, GraphACL (Xiao et al., 2023) captures 1-hop local neighborhood information and two-hop monophily similarity without augmentation. S3GCL (Wan et al., 2024) generates low-pass and high-pass biased views using cosine parameterized Chebyshev polynomial filters.

**Contrastive Methods** The contrastive method in GRL comprises two components, contrastive views and contrastive objectives. Contrastive views determine the structural and semantic levels the model can capture, while contrastive objectives determine how the model measures sample similarity and carries out optimization (Zhao et al., 2025). Contrastive objectives primarily focus on comparisons at the node-node and node-graph levels (Xie et al., 2022). At the node–graph level, methods commonly optimize local–global mutual information using the Bayes–Shannon lower bound or the Jensen–Shannon divergence. For instance, DGI (Veličković et al., 2019) maximizes the mutual information between the node embeddings of the original graph and the global graph summary and uses the corrupted graph as the negative sample. MVGRL (Hassani & Khasahmadi, 2020) performs multi-view contrast by comparing two sets of node-level views with the global graph summary in order to maximize inter-view mutual information. Node-node level contrast typically uses noise contrastive estimation (InfoNCE) or normalized temperature-scaled cross-entropy (NT-Xent) loss to bring positive sample pairs closer in the embedding space while pushing negative sample pairs farther apart. For example, NCLA (Shen et al., 2023) introduces a neighbor contrastive loss that regards the anchor and its neighbor nodes across different augmented node views as positive pairs, and all other nodes as negative pairs. GTCA (Liang et al., 2025) proposes a multi-positive sample contrastive loss that uses the intersection of k nearest neighbor sets from multiple views as positive samples, treating all remaining samples as negative samples.

# 3 PRELIMINARIES

Let $\mathcal{G} = (\mathcal{V}, \mathcal{E})$ denote a graph, where $\mathcal{V} = \{v_1, \cdots, v_N\}$, $\mathcal{E} \subseteq \mathcal{V} \times \mathcal{V}$ represent the node set and the edge set respectively. $\boldsymbol{X} \in \mathbb{R}^{N \times F}$ and $\boldsymbol{A} \in \{0, 1\}^{N \times N}$ denote the node feature matrix and the symmetric adjacency matrix, where $\boldsymbol{x}_i \in \mathbb{R}^F$ is the feature vector of $v_i$ and $\boldsymbol{A}_{ij} = 1$ iff $(v_i, v_j) \in \mathcal{E}$, otherwise $\boldsymbol{A}_{ij} = 0$. $\mathcal{N}_i$ represents the first-order neighbors of node $i$ in the graph. $\mathcal{S}_i = \mathcal{N}_i \cup \{v_i\}$ denotes the subgraph centered on node $i$ and containing its 1-hop neighbors. Given $\boldsymbol{X}$ and $\boldsymbol{A}$ as the input, the proposed model employs the GNN encoder $f(\boldsymbol{X}, \boldsymbol{A})$ to learn the representations of nodes $\boldsymbol{H} = f(\boldsymbol{X}, \boldsymbol{A}) \in \mathbb{R}^{N \times F'}$, $F' \ll F$. The aggregation graph is generated by aggregating the decoupled views. These views are learned by optimizing the joint contrastive loss, without access to the labels of downstream tasks.

# 4 METHODOLOGY

In this section, we describe the MGDRL in detail from three perspectives, including multi-view graph disentanglement, fuzzy self-attention aggregation and subgraph contrastive learning. Finally, we analyze the time complexity of MGDRL. Figure 1 shows the overall architecture of MGDRL.

## 4.1 MULTI-VIEW GRAPH DISENTANGLEMENT

Graph disentanglement can promote the distinctiveness and diversity of the decoupled views, without relying on any prior knowledge or manually defined data augmentation strategies. In MGDRL, we employ multi-view graph disentanglement (MGD) to generate three decoupled views for all benchmark datasets. Based on their respective functions described in the Section 4.2, we designate

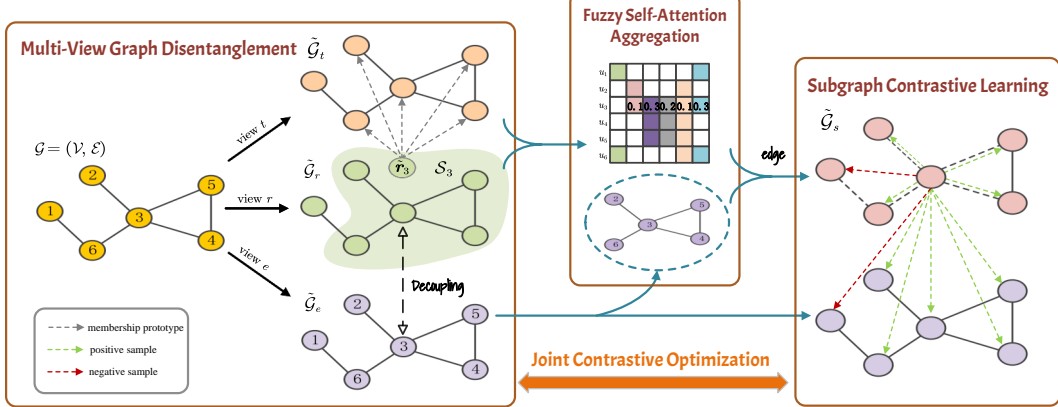

Figure 1: The overall architecture of MGDRL. The original graph generates three decoupled views $t$, $r$ and $e$ via MGD. For each subgraph $\mathcal{S}_i$, the fuzzy self-attention weights are computed from the fuzzy membership between the local readout of view $r$ and the subgraph's nodes in view $t$. Then, the fuzzy self-attention aggregation is performed on view $e$ to produce aggregation graph ($s$) and the subgraph contrastive learning is conducted between view $e$ and view $s$. Finally, we employ MGD and subgraph contrastive learning to jointly optimize the aggregation graph and decoupled views.

the three decoupled views as *transition* ($t$), *readout* ($r$) and *embedding* ($e$). For each decoupled view $z \in \{t, r, e\}$, the edge coefficient between two connected nodes, say $v_i$ and $v_j$, can be learned as

$$\alpha_{ij}^z = \frac{\exp(\text{LeakyReLU}_\rho(\boldsymbol{a}_z[\boldsymbol{W}_z\boldsymbol{x}_i\|\boldsymbol{W}_z\boldsymbol{x}_j]))}{\sum_{v_p \in \mathcal{S}_i} \exp(\text{LeakyReLU}_\rho(\boldsymbol{a}_z[\boldsymbol{W}_z\boldsymbol{x}_i\|\boldsymbol{W}_z\boldsymbol{x}_p]))}, \tag{1}$$

where $\alpha_{ij}^z = 0$ if $\boldsymbol{A}_{ij} = 0$, $\boldsymbol{a}_z \in \mathbb{R}^{2F'}$ is the learnable attention vector of the head $z$, $\boldsymbol{W}_z \in \mathbb{R}^{F' \times F}$ is the learnable weight matrix of the head $z$ which maps each input node feature $\boldsymbol{x}_i \in \mathbb{R}^F$ to an $F'$-dimensional hidden representation, $\|$ is the concatenation operation, and $\text{LeakyReLU}_\rho(\cdot)$ is the LeakyReLU nonlinearity with negative slope $\rho$. Then, representation diffusion over the adjacency matrix is employed to capture the local readout of view $r$, as

$$\widetilde{\boldsymbol{R}} = \widehat{\boldsymbol{D}}^{-1}\,\widehat{\boldsymbol{A}}\,\boldsymbol{H}_r \quad, \tag{2}$$

where $\widetilde{\boldsymbol{R}} \in \mathbb{R}^{N \times F'}$ is the local readout matrix of the view $r$, $\boldsymbol{H}_r$ is the embedding matrix of view $r$, $\widehat{\boldsymbol{A}} = \boldsymbol{A} + \boldsymbol{I}_N \in \mathbb{R}^{N \times N}$ is the adjacency matrix augmented with self-loops, $\widehat{\boldsymbol{D}}_{i,i} = \sum_{j=1}^N \widehat{\boldsymbol{A}}_{i,j}$ is the corresponding degree matrix.

**Graph disentanglement loss.** Using the local readout $\widetilde{\boldsymbol{r}}$ as an anchor, we simultaneously maximize its similarity with the embedding of view $r$ and minimize its similarity with the embedding of view $e$. This contrasting constraint forces the embeddings of view $r$ and view $e$ to separate in the vector space, thereby reducing their inter-view correlation and achieving graph disentanglement. The graph disentanglement loss is defined as

$$\mathcal{L}_d = -\frac{1}{N}\sum_{i=1}^N \Big[\log \sigma\big(\varphi(\widetilde{\boldsymbol{r}}_i, \boldsymbol{h}_i^r)\big) \,+\, \log\big(1 - \sigma(\varphi(\widetilde{\boldsymbol{r}}_i, \boldsymbol{h}_i^e))\big)\Big], \tag{3}$$

where $\widetilde{\boldsymbol{r}}_i$ is the local readout for node $i$ of the decoupled view $r$, $\varphi(\boldsymbol{u}, \boldsymbol{v}) = \boldsymbol{u}^\top \boldsymbol{W}_d \boldsymbol{v}$ is a bilinear discriminator with parameters $\boldsymbol{W}_d \in \mathbb{R}^{F' \times F'}$, $\sigma(\cdot)$ is a sigmoid function. $\boldsymbol{h}_i^r$ and $\boldsymbol{h}_i^e$ are the embeddings of node $i$ in view $r$ and view $e$, respectively.

## 4.2 FUZZY SELF-ATTENTION AGGREGATION

After obtaining the decoupled views, we aggregate these views into an aggregation graph, which serves as a bridge for the joint contrastive optimization of MGD and subgraph contrastive learning (described in Section 4.3). Considering the fuzzy and overlapping nature of community structure in

graphs, we adopt local membership degrees as fuzzy self-attention weights to aggregate the decoupled views into the aggregation graph. In this approach, we assign three decoupled views distinct roles: view $t$ serves as the foundational view to obtain basic features, while view $r$ provides local readout for computing fuzzy self-attention weights. The node embeddings of view $e$ are then aggregated using the fuzzy self-attention weights to yield the embeddings of the aggregation graph.

Specifically, in view $t$, each node $j \in \mathcal{S}_i$ acts as a membership prototype, and the local readout vector $\widetilde{\boldsymbol{r}}_i$ serves as the query to be assigned. The fuzzy self-attention weight is then defined as

$$u_{ij} = \frac{\left\|\widetilde{\boldsymbol{r}}_i - \boldsymbol{h}_j^t\right\|_2^{-\frac{2}{m-1}}}{\displaystyle\sum_{k \in \mathcal{S}_{(i)}} \left\|\widetilde{\boldsymbol{r}}_i - \boldsymbol{h}_k^t\right\|_2^{-\frac{2}{m-1}}}, \tag{4}$$

where $\sum_{j \in \mathcal{S}_i} u_{ij} = 1$, $\|\cdot\|_2$ is the $L^2$ norm, $m$ is the fuzziness weighting exponent that controls the fuzziness degree of clustering outcomes.

Then, we aggregate the node embeddings in view $e$ by the fuzzy self-attention weights to obtain the embeddings of aggregation graph, which we denote as $s$. Although there are no explicit edge relationships between the embeddings of aggregation graph, the homophily principle of networks suggests that similar subgraphs are closely related (McPherson et al., 2001). In MGDRL, we consider subgraph pairs with intersections as "closely connected subgraphs" and establish an edge between their embedding pairs. The embedding for each subgraph is obtained as

$$\boldsymbol{h}_i^s = \sum_{j \in \mathcal{S}_{(i)}} u_{ij}\, \boldsymbol{h}_j^e. \tag{5}$$

The edge relationship of the aggregation graph is defined as $\tilde{\mathcal{E}}_s = \{(\boldsymbol{h}_i^s, \boldsymbol{h}_j^s) \mid \mathcal{S}_i \cap \mathcal{S}_j \neq \emptyset, i \neq j\}$, and the set of subgraphs that intersect with subgraph $i$ is defined as $\mathcal{P}_i = \{\mathcal{S}_j \mid \mathcal{S}_i \cap \mathcal{S}_j \neq \emptyset, i \neq j\}$. Since $\mathcal{S}_i$ is a 1-hop neighborhood subgraph for node $i$, its adjacency matrix coincides exactly with that of the original graph $\mathcal{G}$, i.e., $\tilde{\boldsymbol{A}}_s = \boldsymbol{A} \in \{0,1\}^{N \times N}$. Then, the aggregation graph can be defined as $\tilde{\mathcal{G}}_s = (\boldsymbol{H}_s, \tilde{\boldsymbol{A}}_s, \tilde{\mathcal{E}}_s)$.

### 4.3 Subgraph Contrastive Learning

Our experiments indicate that a single graph disentanglement loss does not provide sufficient guidance for the model to generate meaningful features. Therefore, it is necessary to combine disentanglement with other pretext tasks to constrain the correlations among the decoupled views. To address these issues, we propose subgraph contrastive learning. Traditional node-node and node-graph level contrastive learning methods suffer from limitations such as node-level noise and the lack of localized structural features. The proposed subgraph contrastive learning extends the contrastive objective to the subgraph-level and achieves a integration between local and global information.

**Subgraph contrastive loss.** Subgraph contrastive learning uses subgraph-level and node-level views for intra-view and inter-view comparison. Therefore, the aggregation graph embedding $\boldsymbol{h}_i^s$ derives its positive samples from two sources:

- $\{\boldsymbol{h}_j^s \mid \mathcal{S}_j \in \mathcal{P}_i\}$, the aggregation graph embedding of the subgraph $j$ that intersects subgraph $i$.
- $\{\boldsymbol{h}_k^e \mid v_k \in \mathcal{S}_i\}$, the node embedding of the node $k$ contained in subgraph $i$.

Then, the intra-view subgraph contrastive loss, which can be regarded as subgraph-subgraph level comparison, can be formulated as

$$\ell_{intra}(\boldsymbol{h}_i^s) = -\log \frac{\sum_{v_k \in \mathcal{N}_i} \exp\big(\theta(\boldsymbol{h}_i^s, \boldsymbol{h}_k^s)/\tau\big)}{\sum_{i \neq j} \exp\big(\theta(\boldsymbol{h}_i^s, \boldsymbol{h}_j^s)/\tau\big)}, \tag{6}$$

and the inter-view subgraph contrastive loss, which can be regarded as node-subgraph level comparison, can be formulated as

$$\ell_{inter}(\boldsymbol{h}_i^s) = -\log \frac{\sum_{v_k \in \mathcal{S}_i} \exp\big(\theta(\boldsymbol{h}_i^s, \boldsymbol{h}_k^e)/\tau\big)}{\sum_j \exp\big(\theta(\boldsymbol{h}_i^s, \boldsymbol{h}_j^e)/\tau\big)}, \tag{7}$$

where $\tau$ is a temperature parameter, $\theta(\cdot)$ is the cosine similarity. The terms in the denominator of Eq. (6) and Eq. (7) can be decomposed as

$$\sum_{i \neq j} \exp(\theta(\boldsymbol{h}_i^s, \boldsymbol{h}_j^s)/\tau) = \underbrace{\sum_{\mathcal{S}_j \in \mathcal{P}_i} \exp(\theta(\boldsymbol{h}_i^s, \boldsymbol{h}_j^s)/\tau)}_{intra-view\ pos} + \underbrace{\sum_{\mathcal{S}_j \notin \mathcal{P}_i} \exp(\theta(\boldsymbol{h}_i^s, \boldsymbol{h}_j^s)/\tau)}_{intra-view\ neg},$$

$$\sum_{j} \exp(\theta(\boldsymbol{h}_i^s, \boldsymbol{h}_j^e)/\tau) = \underbrace{\sum_{v_j \in \mathcal{S}_i} \exp(\theta(\boldsymbol{h}_i^s, \boldsymbol{h}_j^e)/\tau)}_{inter-view\ pos} + \underbrace{\sum_{v_j \notin \mathcal{S}_i} \exp(\theta(\boldsymbol{h}_i^s, \boldsymbol{h}_j^e)/\tau)}_{inter-view\ neg},$$

where the non-connected aggregation graph embeddings and non-containing nodes of subgraph $i$ are regarded as negative pairs, respectively. Minimizing Eq. (6) and Eq. (7) would maximize the agreement between positive pairs and minimize that of negative pairs. The final subgraph contrastive loss is defined as

$$\mathcal{L}_s = \frac{1}{N} \sum_{i=1}^{N} \left( \ell_{intra} \left( \boldsymbol{h}_i^s \right) + \ell_{inter} \left( \boldsymbol{h}_i^s \right) \right). \tag{8}$$

**Joint contrastive optimization.** Joint contrastive optimization is a dynamic and coordinated process between MGD and subgraph contrastive learning. Connected via fuzzy self-attention aggregation, MGD produces distinct and diverse decoupled views, while subgraph contrastive learning simultaneously constrains MGD and enables these views to integrate both local and global information. The final joint contrastive loss is defined as

$$\mathcal{L}_J = \mathcal{L}_d + \lambda \cdot \mathcal{L}_s \quad , \tag{9}$$

where $\lambda$ is a tunable constraint weight parameter.

### 4.4 TIME COMPLEXITY

For MGD to produce $K$ views, the time complexity is $\mathcal{O}((NFF' + |\mathcal{E}|F')K)$, where $N$ and $|\mathcal{E}|$ are the number of nodes and edges in graph $\mathcal{G}$, with $F$ and $F'$ denoting the input feature dimension and the output embedding dimension. The time complexity of the graph disentanglement loss is $\mathcal{O}(NF'^2 + |\mathcal{E}|F')$. Let $M = \frac{|\mathcal{E}|}{N}$ denote the average node degree, the time complexity of fuzzy self-attention aggregation is $\mathcal{O}(NMF')$ and the time complexity of subgraph contrastive learning is $\mathcal{O}(N(M+N)F')$. Thus, the time complexity of MGDRL is $\mathcal{O}(NFF'K + |\mathcal{E}|F'(K+1) + NF'^2 + N(M+N)F')$. Since $|\mathcal{E}| \ll N^2$ and $M \ll F' \ll F$, the overall time complexity of MGDRL is $\mathcal{O}(NFF'K + N^2F')$. Note $K$ is very small (e.g., 3) in our experiments, so the time complexity of MGDRL is comparable to the representative node-node GRL methods, e.g., GRACE (Zhu et al., 2020).

## 5 EXPERIMENTS

### 5.1 DATASETS

In our experiments, we evaluate our method on seven widely-used datasets for semi-supervised node classification, including three citation networks, i.e., Cora, Citeseer, Pubmed (Sen et al., 2008), a reference network constructed based on Wikipedia, i.e., Wiki-CS (Mernyei & Cangea, 2020), a co-authorship network, i.e., Coauthor-CS (Shchur et al., 2018), and two product co-purchase networks, i.e., Amazon-Computers and Amazon-Photo (Shchur et al., 2018). Datasets Cora, Citeseer and Pubmed were also used to evaluate performance on the node clustering task.

### 5.2 BASELINES

We thoroughly consider 20 state-of-the-art methods for comparison on semi-supervised node classification and node clustering tasks. Baselines trained with labels: GCN (Kipf & Welling, 2017), GAT (Velickovic et al., 2017), CGPN (Wan et al., 2021b), CG3 (Wan et al., 2021a). Baselines trained without labels: DGI (Veličković et al., 2019), GMI (Peng et al., 2020), MVGRL (Hassani & Khasahmadi, 2020), GRACE (Zhu et al., 2020), GCA (Zhu et al., 2021), ProGCL (Xia et al., 2021),

Table 1: Node classification performance. $X, A, Y$ denote the node attributes, adjacency matrix, and labels in the datasets. $S, E$ denote the diffusion matrix and edge feature matrix. OOM signifies out-of-memory.

| Methods | Available Data | Datasets | | | | | | |
|---------|----------------|----------|----------|----------|----------|----------|----------|----------|
| | | Cora | Citeseer | Pubmed | CS | Photo | Computers | WikiCS |
| GCN | $X, A, Y$ | 79.6±1.8 | 66.0±1.2 | 79.0±2.5 | 90.0±0.6 | 86.3±1.6 | 76.4±1.8 | 67.3±1.5 |
| GAT | $X, A, Y$ | 81.2±1.6 | 68.9±1.8 | 78.5±1.8 | 90.9±0.7 | 86.5±2.1 | 77.9±1.8 | 68.6±1.9 |
| CGPN | $X, A, Y$ | 74.0±1.7 | 63.7±1.6 | 73.3±2.5 | 83.5±1.4 | 84.1±1.5 | 74.7±1.3 | 66.1±2.1 |
| CG3 | $X, A, Y$ | 80.6±1.6 | 70.9±1.5 | 78.9±2.6 | 90.6±1.0 | 89.4±1.9 | 77.8±1.7 | 68.0±1.5 |
| DGI | $X, A$ | 82.1±1.3 | 71.6±1.2 | 78.3±2.4 | 92.0±0.5 | 83.5±1.2 | 78.8±1.1 | 69.1±1.4 |
| GMI | $X, A$ | 79.4±1.2 | 66.9±2.2 | 76.8±2.3 | 88.5±0.8 | 86.7±1.5 | 76.1±1.2 | 67.8±1.8 |
| MVGRL | $X, S, A$ | 82.4±1.5 | 71.1±1.4 | 79.5±2.2 | 91.5±0.6 | 89.7±1.2 | 78.7±1.7 | 69.2±1.2 |
| GRACE | $X, A$ | 79.6±1.4 | 67.0±1.7 | 74.6±3.5 | 90.0±0.7 | 87.9±1.4 | 76.8±1.7 | 67.8±1.4 |
| GCA | $X, A$ | 79.0±1.4 | 65.6±2.4 | 81.5±2.5 | 90.9±1.1 | 87.0±1.9 | 76.9±1.4 | 67.6±1.3 |
| AFGRL | $X, A$ | 78.6±1.3 | 70.8±2.1 | 76.4±2.5 | 91.4±0.6 | 89.2±1.1 | 77.7±1.1 | 68.0±1.7 |
| SUGRL | $X, A$ | 81.3±1.2 | 71.0±1.8 | 80.5±1.6 | 91.2±0.9 | 90.5±1.9 | 78.2±1.2 | 68.7±1.1 |
| ARIEL | $X, A$ | 81.3±1.3 | 70.9±1.4 | 74.2±2.5 | 90.2±0.9 | 90.6±1.8 | 81.3±1.4 | 70.5±2.1 |
| NCLA | $X, A$ | 82.2±1.6 | 71.7±0.9 | 82.0±1.4 | 91.5±0.7 | 90.2±1.3 | 79.8±1.5 | 70.3±1.7 |
| GraphACL | $X, A$ | 82.0±1.1 | 71.5±1.4 | 78.6±1.9 | 86.9±1.2 | 90.0±1.0 | - | - |
| PiGCL | $X, A$ | 80.0±1.5 | 71.2±1.1 | 76.5±3.5 | 91.0±0.7 | 71.8±3.4 | - | - |
| AFECL | $X, E, A$ | 82.1±1.3 | 71.3±1.3 | 81.2±1.7 | 90.9±1.3 | 89.2±1.2 | - | - |
| GTCA | $X, A$ | 82.5±1.3 | 68.3±1.4 | OOM | **92.5±0.6** | 90.5±1.2 | 79.2±1.4 | 69.7±1.5 |
| MGDRL(Ours) | $X, A$ | **83.1±1.4** | **72.3±1.1** | 82.0±1.2 | 92.0±0.3 | **91.4±0.9** | **81.4±1.4** | **73.7±1.9** |

Table 2: Node clustering performance. OOM signifies out-of-memory.

| Method | Cora | | Citeseer | | Pubmed | |
|--------|------|------|----------|------|--------|------|
| | NMI | ARI | NMI | ARI | NMI | ARI |
| DGI | 0.5370 | 0.4469 | 0.4185 | 0.4140 | 0.3188 | 0.3165 |
| GRACE | 0.4758 | 0.3633 | 0.3960 | 0.3977 | 0.3508 | 0.3286 |
| GCA | 0.4510 | 0.3104 | 0.3737 | 0.3675 | 0.3307 | 0.2919 |
| ProGCL | 0.5131 | 0.3434 | 0.4115 | 0.4219 | OOM | OOM |
| Local-GCL | 0.5386 | 0.4479 | 0.4508 | 0.4494 | 0.3469 | 0.3304 |
| AFGRL | 0.3525 | 0.2465 | 0.3896 | 0.3958 | 0.3689 | 0.2474 |
| SUGRL | 0.2977 | 0.2766 | 0.4454 | 0.4507 | 0.2977 | 0.2766 |
| NCLA | 0.6089 | 0.5750 | 0.4553 | 0.4610 | 0.2523 | 0.2383 |
| PiGCL | 0.5494 | 0.4670 | 0.4581 | 0.4720 | **0.3784** | **0.3612** |
| UniFilter | 0.5212 | 0.4744 | 0.4422 | 0.4318 | 0.3145 | 0.2773 |
| GTCA | 0.5588 | 0.5063 | 0.3392 | 0.3125 | OOM | OOM |
| MGDRL(Ours) | **0.6190** | **0.6076** | **0.4594** | **0.4732** | 0.3643 | 0.3459 |

Local-GCL (Zhang et al., 2022), AFGRL (Lee et al., 2022), SUGRL (Mo et al., 2022), ARIEL (Feng et al., 2022), NCLA (Shen et al., 2023), GraphACL (Xiao et al., 2023), PiGCL (He et al., 2024), UniFilter (Huang et al., 2024), GTCA (Liang et al., 2025), AFECL (Li et al., 2025).

## 5.3 EXPERIMENTAL SETTINGS

The proposed MGDRL was implemented using PyTorch 2.5.1 (Paszke et al., 2019) and Deep Graph Library 2.0.0 (Wang et al., 2019), and trained by the Adam optimizer on all datasets. For the node classification task, we allow GRL baselines and MGDRL to learn embeddings in an unsupervised manner, then use these embeddings to train and test a $L_2$-regularized logistic regression (LR) classifier for semi-supervised node classification. For Cora, Citeseer and Pubmed, we followed (Yang et al., 2016) to randomly select 20 nodes per class for training, 500 nodes for validation and the remaining nodes for test. For Coauthor-CS and Amazon-Photo, we followed Liu et al. (2020) to randomly select 20 nodes per class for training, 30 nodes per class for validation, and the remaining nodes for testing. Note that for the GRL baselines and MGDRL, which learn embeddings from unlabeled data, the validation set was just used to tune the hyperparameters of the LR classifier, rather than the GRL models. For each dataset, we conducted 20 random splits of training/validation/test, and reported the averaged performance of all algorithms on the same random splits. For the node

Table 3: Ablation study of MGDRL for node classification on all seven datasets.

| Variants | $\mathcal{L}_d$ | $\mathcal{L}_s$ | $\mathcal{L}_i$ | Cora | Citeseer | Pubmed | CS | Photo | Computers | WikiCS |
|---|---|---|---|---|---|---|---|---|---|---|
| Raw data | | | | 63.0±1.7 | 51.5±1.9 | 72.5±2.2 | 87.8±0.7 | 87.7±1.5 | 75.8±1.7 | 70.6±2.4 |
| Variant 1 | ✔ | | | 64.8±1.9 | 56.0±2.0 | 72.6±2.6 | 84.7±1.1 | 84.1±2.2 | 73.6±1.3 | 71.8±1.4 |
| Variant 2 | | ✔ | | 81.8±1.3 | 70.4±1.5 | 80.2±1.9 | 90.3±0.8 | 89.9±2.2 | 80.6±1.5 | 71.6±2.0 |
| Variant 3 | | | ✔ | 63.3±2.1 | 66.2±2.2 | 71.8±2.0 | 77.5±1.3 | 78.9±1.5 | 69.4±2.1 | 63.5±1.8 |
| Variant 4 | ✔ | | ✔ | 76.4±1.5 | 68.8±1.5 | 71.3±2.3 | 80.1±1.3 | 81.5±1.9 | 65.6±2.2 | 59.7±2.1 |
| MGDRL | ✔ | ✔ | | **83.1±1.4** | **72.3±1.1** | **82.0±1.2** | **92.0±0.3** | **91.4±0.9** | **81.4±1.4** | **73.7±1.9** |

clustering task, we followed He et al. (2024) to directly input the obtained embedding into a randomly initialized K-Means predictor with up to 500 iterations. We ran this process 10 times and reported the average NMI and ARI.

## 5.4 OVERALL PERFORMANCE

**Node Classification.** Table 1 presents the node classification accuracy of MGDRL on seven benchmark datasets. These node classification results demonstrate MGDRL's robust performance across all seven datasets and validate the superiority of MGD with joint contrastive optimization in feature extraction. Compared to supervised GNNs, our method achieves an average improvement of 3.9% over GCN and GAT. When evaluated against other supervised methods like CGPN and CG3, MGDRL comprehensively outperforms them with gains of up to 9.1%. In comparison with self-supervised GRL methods that employ graph augmentation and node- and graph-level contrastive learning, such as GRACE, NCLA and PiGCL, our approach achieves comprehensive leading performance. MGDRL also maintains advantages over other methods without graph augmentation, including GraphACL, AFECL and GTCA.

**Node Clustering.** Table 2 shows the node clustering performance of MGDRL on the Cora, Citeseer and Pubmed datasets. MGDRL demonstrates strong clustering performance on the Cora and Citeseer datasets, achieving average improvements of 10.33% in NMI and 15.99% in ARI over the baseline GRACE. MGDRL also achieved the third-best NMI and second-best ARI performance on Pubmed. Compared to methods without graph augmentation such as GTCA, MGDRL utilizes joint contrastive learning to enable the representations to integrate local and global information, thereby facilitating the formation of distinct clusters for nodes with different labels.

## 5.5 ABLATION STUDY

Table 3 presents the ablation study results of MGDRL on all seven datasets. "Raw data" denotes the results obtained by directly processing the raw graph data through GAT without any modifications. Variants 1, 2, and MGDRL demonstrate the impact of MGD ($\mathcal{L}_d$) and subgraph contrastive learning ($\mathcal{L}_s$) in the joint contrastive optimization. The performance of Variant 1 is highly unstable, with accuracy significantly decreasing in three datasets compared to the raw data, indicating that MGD alone fails to effectively capture meaningful representations. To more intuitively illustrate the constraining effect of subgraph contrastive learning on MGD, we introduced InfoNCE ($\mathcal{L}_i$) as a control, resulting in Variants 3 and 4. From Variants 1, 3, and 4, it can be observed that traditional contrastive learning methods cannot provide effective driving force for MGD, instead, they may even lead to mutual performance degradation. The ablation results collectively demonstrate the effectiveness of the joint contrastive optimization.

## 5.6 VISUALIZATION AND HYPERPARAMETER ANALYSIS

**Visualization.** We use t-SNE (Van der Maaten & Hinton, 2008) for visualization to more intuitively show the embedding distributions obtained by variants of MGDRL and four other baseline methods on Cora, as shown in Figure 2. The visualization of Variant 1 also indicates that using MGD alone leads the model to learn largely uninformative representations. The comparison between Variants 1 and 2 reveals how MGD and subgraph contrastive learning produce different representations: MGD encourages views to capture disentangled information, resulting in widely dispersed node

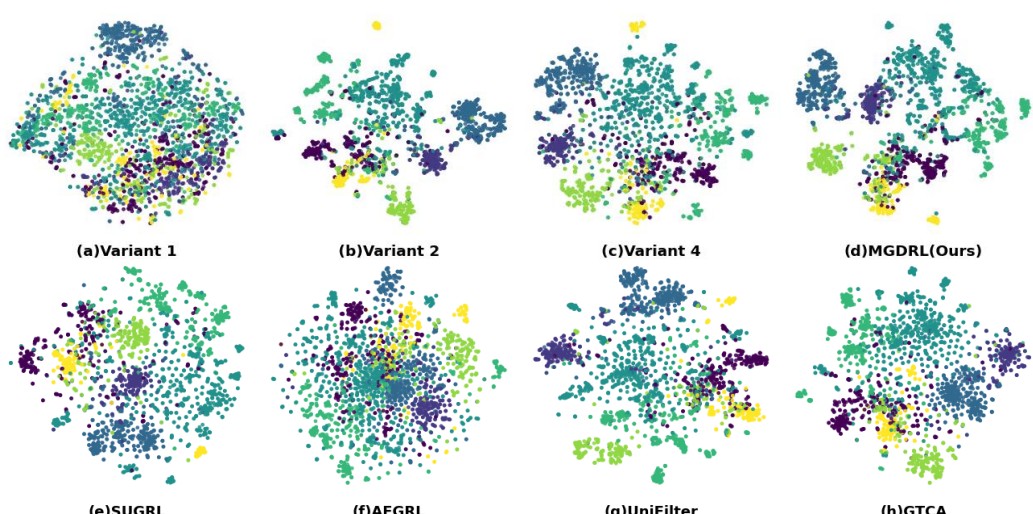

Figure 2: Visualization of variants and four baseline GRL embeddings on Cora with t-SNE.

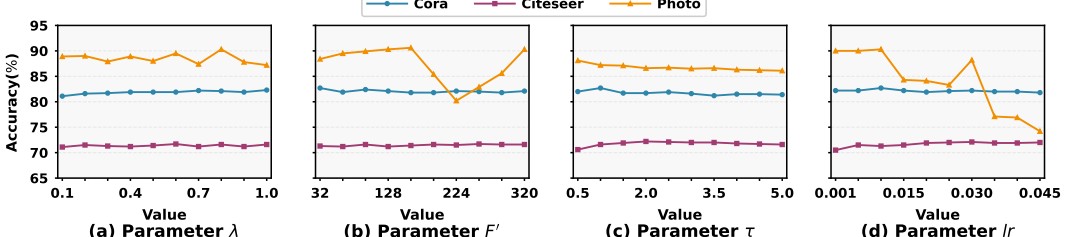

Figure 3: Sensitivity analysis of the hyperparameters $\lambda$, $F'$, $\tau$ and $lr$ on MGDRL.

embeddings, whereas subgraph contrastive learning encourages nodes to integrate local and global information, thereby forming cluster structures. Compared with Variant 4 and the other baselines, MGDRL both pushes nodes of different classes farther apart and preserves cluster structure, which is a consequence of the joint contrastive optimization.

**Hyperparameter analysis.** We conducted sensitivity analysis on Cora, Citeseer, and Photo across four parameters: constraint weight $\lambda$, hidden layer dimension $F'$, temperature parameter $\tau$, and learning rate $lr$. The results in Figure 3 demonstrate that MGDRL exhibits highly stable performance on Cora and Citeseer. For the Photo dataset, MGDRL continues to perform well with respect to $\lambda$ and $\tau$. However, performance degradation occurs when using larger values for $F'$ and $lr$.

## 6 CONCLUSION

Existing GRL methods employ graph augmentation to construct views and utilize node- and graph-level contrastive learning for optimization. However, artificial perturbations result in the loss of critical information in graph data, while node- and graph-level contrastive learning suffers from noise sensitivity and an inability to perceive local structure. To address these issues, we propose MGDRL, a graph representation learning framework that performs MGD via joint contrastive optimization. MGD generates decoupled views, which are used to generate the aggregation graph by fuzzy self-attention aggregation. Then, the decoupled view (*embedding*) and the aggregation graph are used for node-subgraph and subgraph-subgraph level comparisons in subgraph contrastive learning. By jointly optimizing the MGD and subgraph contrastive learning, the decoupled views learn distinct and diverse representations, while the aggregation graph integrates both local and global information. Experiments on benchmark datasets demonstrate the effectiveness of MGDRL.

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
