# OpenReview forum: "Multi-View Graph Disentanglement via Joint Contrastive Optimization"
_ICLR.cc/2026/Conference — ICLR 2026 Conference Withdrawn Submission_

### Official Review · Reviewer_iXNe · 2025-10-26

**Soundness:** 2
**Presentation:** 3
**Contribution:** 2
**Rating:** 4
**Confidence:** 5

**Summary:**

The paper addresses limitations in graph representation learning, where existing methods rely on graph augmentation and contrastive learning, potentially leading to information loss and view homogeneity. The authors propose MGDRL, a framework that employs multi-view graph disentanglement to generate decoupled views (transition, readout, and embedding), fuzzy self-attention aggregation to create an aggregation graph, and subgraph contrastive learning for joint optimization. The key contribution is the joint contrastive optimization that promotes view divergence while integrating local and global information. Experimental results on benchmark datasets show superior performance in semi-supervised node classification and node clustering tasks compared to baselines.

**Strengths:**

1.	The proposed MGD and joint contrastive optimization offer a novel perspective by promoting view divergence without relying on graph augmentation.
2.	The paper is easy to follow with clear descriptions.

**Weaknesses:**

1.	In intro, the authors discussed the over-smoothing caused by deep GNNs. However, there is no further discussion on this point, and it seems that the proposed model has nothing to do with this problem. So, How does the proposed model perform when the network is deep? Or, How can the proposed model overcome oversmoothing issue?
2.	The time complexity analysis in Section 4.4 is provided but not compared empirically to baselines in terms of training time or resource usage.
3.	Why do the authors only perform node clustering on citeseer, cora and pubmed?
4.	There is no experimental convergence analysis for subgraph contrastive loss and graph disentanglement loss.
5.	Lacking comparison to the following important related work on disentangled graph contrastive learning:

    a.	Li, Haoyang, et al. "Disentangled contrastive learning on graphs." Advances in Neural Information Processing Systems 34 (2021): 21872-21884.

**Questions:**

See weaknesses.

---

### Official Review · Reviewer_hN9x · 2025-10-31

**Soundness:** 3
**Presentation:** 2
**Contribution:** 3
**Rating:** 4
**Confidence:** 4

**Summary:**

The paper proposes MGDRL, an augmentation-free self-supervised method that disentangles a graph into three learnable views (transition, readout, embedding) and uses fuzzy self-attention to aggregate subgraph embeddings, trained with a joint loss that combines a disentanglement term and subgraph-level contrast; it achieves strong results on standard node-classification and clustering benchmarks.

**Strengths:**

A clear replacement of brittle data augmentations with learned multi-view disentanglement.

A principled fuzzy-membership mechanism that yields interpretable subgraph aggregation.

Broad empirical coverage with ablations showing the joint objective is necessary.

**Weaknesses:**

Limited diagnostics proving real disentanglement (no inter-view CKA/HSIC).

Fixed 1-hop subgraph design that may struggle on heterophilous or long-range graphs, incomplete sensitivity on the fuzziness exponent and neighborhood size, and unclear scalability/negative-sampling costs on very large graphs.

It would be better to add inter-view dependence metrics during training, vary or learn the subgraph radius and test on heterophilous graphs, report robustness to the fuzziness exponent and sampling strategies with wall-clock and memory, include stronger subgraph-contrast baselines and parameter/time reports, visualize membership distributions and provide error analyses, and compare linear-probe versus full fine-tuning.

**Questions:**

See above.

---

### Official Review · Reviewer_bZMg · 2025-11-01

**Soundness:** 2
**Presentation:** 2
**Contribution:** 2
**Rating:** 4
**Confidence:** 4

**Summary:**

The paper ‘Multi-View Graph Disentanglement via Joint Contrastive Optimization’ proposes an augmentation-free framework that overcomes the limitations of conventional graph representation learning relying on handcrafted augmentations. By jointly performing multi-view graph disentanglement and subgraph-level contrastive optimization, the method captures diverse structural, local, and semantic information. Through fuzzy self-attention aggregation, it introduces an augmentation-free, multi-view contrastive framework for graph representation learning that disentangles structural, local, and semantic information, then harmonizes them via fuzzy aggregation and joint contrastive optimization, leading to more robust and interpretable graph embeddings.

**Strengths:**

1.This paper introduces a framework(MGDRL), which departs from conventional graph contrastive learning paradigms that rely heavily on manual graph augmentations. Instead, it achieves view diversification through a disentanglement mechanism, thus avoiding information distortion caused by artificial perturbations.

2.The work formulates joint contrastive optimization as a new learning objective that unifies disentanglement and contrastive processes—an innovative formulation that removes a key limitation of prior augmentation-based GRL methods.

3.It addresses long-standing issues of view redundancy and noise sensitivity, introducing a generalizable framework that can be adapted to a wide range of graph-based tasks. The demonstrated improvements on both node classification and clustering tasks, together with the theoretical soundness of the proposed optimization mechanism, underscore its potential impact on advancing the robustness and interpretability of future graph representation models.

**Weaknesses:**

1.Insufficient exploration of model generality across graph domains.The experiments are conducted exclusively on small to medium-sized citation, co-authorship, and product co-purchase graphs. These datasets are relatively homogeneous and static. The proposed framework’s applicability to more complex graph types—such as temporal, heterogeneous, or large-scale web graphs.

2.Limited interpretability and qualitative evaluation.The visualization (Figure 2) demonstrates cluster separation qualitatively, but the paper does not analyze what semantic or structural aspects each decoupled view captures. Since disentanglement is a central claim, it would be valuable to include interpretability experiments—such as examining which structural motifs, node attributes, or communities are emphasized in each view.

3.Clarity of fuzzy self-attention formulation and hyperparameter sensitivity.Although the fuzzy self-attention mechanism is innovative, its mathematical presentation (Eq. 4) and interaction with the fuzziness exponent m are somewhat opaque.In the hyperparameter analysis of the article, no hyperparameter analysis experiments were conducted on all the parameters mentioned in the text. For instance, the sensitivity analysis of the fuzzy parameter m and the decoupling weight λ in the formula is brief and does not explain how they affect the learning objective.

4.The roles and architectural choices for the transition (t), readout (r) and embedding (e) encoders are described but not rigorously defined—e.g., whether they share weights, their receptive fields, or how their capacities differ. Specify encoder architectures, parameter sharing schemes, and design rationales. Add ablations that vary encoder depth, width, and shared vs. separate weights to show how encoder choices affect disentanglement and downstream performance.

5.Sections 4.2 and 4.3 feel loosely connected: the transition from fuzzy aggregation (Eq. 5) to subgraph contrastive losses (Eq. 6–7) lacks a formal motivation or intermediate derivation. Provide a clearer derivation showing how fuzzy weights produce meaningful subgraph embeddings and why those embeddings are the correct positive/negative units for the proposed contrastive objectives.

6. The current conclusion largely repeats the abstract by restating the main components of MGDRL (MGD, fuzzy self-attention, and subgraph contrastive learning) rather than synthesizing the key findings and insights gained from experiments.The section ends abruptly after restating empirical effectiveness, without outlining possible extensions.

**Questions:**

1.Could the authors provide results or at least preliminary experiments on more diverse and larger graph benchmarks (e.g., OGBN-Papers100M or a heterogeneous molecular/social graph) to demonstrate MGDRL’s scalability and domain generality? Reporting wall-clock training time, peak memory, and any engineering tricks used for large graphs would help assess practical applicability.

2.Do the authors expect fuzzy self-attention and the subgraph contrastive loss to work for temporal or multi-relation graphs? If yes, please either include a short experiment on a temporal or heterogeneous dataset, or explain theoretically how the method adapts to multi-relation dynamics—otherwise clarify the intended scope.

3.Please provide per-view analyses that demonstrate what each decoupled view encodes (e.g., motif/community prevalence, attribute correlation, or role-based statistics). Useful additions would be attention heatmaps per view, per-view mutual information with node labels/attributes, or a metric like attention entropy — such evidence would substantiate the claim of meaningful disentanglement.

4.Eq. (4) and the fuzziness exponent m lack practical guidance. Please (a) give intuition for typical m ranges and their effect on membership sharpness, and (b) include a focused sweep of m and disentanglement weight λ showing training stability, convergence curves, and downstream accuracy—this will clarify sensitivity and reproducibility.

5.The transition from fuzzy aggregation (Eq. 5) to subgraph contrastive losses (Eq. 6–7) is not fully motivated. Can the authors (a) provide a short derivation or lemma showing why the fuzzy-aggregated embeddings are suitable positive/negative units for subgraph contrastive objectives, or (b) supply empirical probes (e.g., correlation between uᵢⱼ and subgraph similarity) to support this design choice?

6.Would the authors consider expanding the Conclusion to explicitly discuss method limitations (e.g., sensitivity to m, untested graph types) and concrete next steps (e.g., adaptive fuzziness, extensions to heterogeneous/temporal graphs)? It would benefit from a concise but forward-looking discussion—for instance, exploring adaptive fuzzy mechanisms, extending MGDRL to large-scale or temporal graphs, or applying the framework to practical downstream tasks such as molecular or social network analysis.

---

### Official Review · Reviewer_Yx7Y · 2025-11-01

**Soundness:** 2
**Presentation:** 3
**Contribution:** 2
**Rating:** 2
**Confidence:** 4

**Summary:**

This paper introduces a multi-view graph representation learning model (MGDRL) that uses joint contrastive learning to separate multi-view graphs. It also uses a fuzzy self-attention mechanism and subgraph-level contrastive learning. The authors believe this method can learn a lot of different information without needing graph augmentation. Experiments show it works well on several standard datasets for classification and clustering.

**Strengths:**

1. The paper clearly identifies the reliance of mainstream Graph Contrastive Learning (GCL) methods on data augmentation, as well as the limitations of conventional contrastive views (i.e., rely on graph augmentation to construct node-level and graph-level view).
2. The paper proposes a subgraph-level contrastive scheme, which may be more robust to noise than node-level contrast and better at preserving diverse information than graph-level contrast.

**Weaknesses:**

1. The notion of "disentanglement" in this paper is ill-defined. While the graph disentanglement loss  L_d (Eq. 3) promotes divergence, it does not guarantee information-theoretic disentanglement, and the authors should provide analysis to suggest it does.
2. The core concept of this paper is the "Subgraph Contrastive Loss." However, the definition of a "subgraph" is presented in a highly confusing manner across both the "Preliminaries" and "Subgraph Contrastive Learning" sections.
3. This paper says it doesn't use augmentation, but it doesn't talk about or compare itself to other papers that also skip augmentation in graph contrastive learning. That's a big hole in the related research section.
4. The paper doesn't look at how stable its Joint Contrastive Optimization setup is or if it might cause representation collapse. This is a big theoretical thing they skipped, and they should talk about it.

**Questions:**

N/A

---

### Official Review · Reviewer_bR1U · 2025-11-01

**Soundness:** 3
**Presentation:** 2
**Contribution:** 1
**Rating:** 2
**Confidence:** 5

**Summary:**

This paper proposes MGDRL, a framework to address the limitations of GCL methods, such as information loss from manual augmentations and the shortcomings of node-level or graph-level contrast. The framework employs Multi-view Graph Disentanglement (MGD) to generate divergent, decoupled views and constrains them using a novel subgraph-level contrastive loss via joint optimization. Extensive experiments demonstrate its effectiveness on node classification and clustering.

**Strengths:**

- The paper proposes a view generation method via Multi-view Graph Disentanglement (MGD), addressing the information-loss limitations of commonly-used manual graph augmentation strategies.
- It extends the contrastive objective to node-subgraph and subgraph-subgraph levels, which is designed to integrate local and global structural information.
- Experiments conducted on several benchmark datasets demonstrate the effectiveness of the proposed method.

**Weaknesses:**

- The motivation is somewhat weak. The critique of "manual graph augmentations" is not fully current, as several methods already use adaptive augmentations [1,2]. Additionally, the claim that existing objectives are "sensitive to noise" or "overlook local structure" is a key premise that requires stronger theoretical or empirical validation.
- The proposed MGD view generation mechanism (Eq. 1) has an operational overlap with the learnable augmentation in NCLA [1]. Both employ multi-head attention to create adaptive views. The primary addition appears to be only the disentanglement loss.
- The "Subgraph Contrastive Loss" (Eqs. 6 and 7) is conceptually similar to established objectives. Several works [1, 3, 4] have already proposed using 1-hop neighbors as positive samples to leverage homophily, making this a relatively mature contrastive strategy.
- A potential concern is the method's scalability. The reported $\mathcal{O}(N^2F')$ complexity may present computational challenges when applying the model to the large-scale, million-level datasets now common in GCL benchmarks.
- The framework is quite sophisticated, requiring the joint optimization of MGD, a fuzzy self-attention aggregation step, and a custom two-part contrastive loss. This architectural complexity, especially when combined with the $\mathcal{O}(N^2F')$ computational bottleneck, may raise considerations for its practical adoption and ease of implementation.

[1] Neighbor Contrastive Learning on Learnable Graph Augmentation. AAAI 2023.
[2] Why Does Dropping Edges Usually Outperform Adding Edges in Graph Contrastive Learning? AAAI 2025.
[3] HomoGCL: Rethinking Homophily in Graph Contrastive Learning. KDD 2023.
[4] Towards Expansive and Adaptive Hard Negative Mining: Graph Contrastive Learning via Subspace Preserving. WWW 2024.

**Questions:**

Please refer to the Weaknesses section.

---

### Note · Authors · 2025-11-30

I have read and agree with the venue's withdrawal policy on behalf of myself and my co-authors.